# Temperature-dependent dynamic disproportionation in LiNiO$_2$

Andrey D. Poletayev [1,2] ✉, Robert J. Green [3,4], Jack E. N. Swallow[1,2], Lijin An[1,2], Leanne Jones[1,2], Grant Harris[3], Peter Bencok[5], Ronny Sutarto [6], Jonathon P. Cottom [2,7,8], Benjamin J. Morgan [2,7], Robert A. House [1,2], Robert S. Weatherup [1,2] ✉ & M. Saiful Islam [1,2,7] ✉

Nickelate materials offer diverse functionalities for energy and computing applications. Lithium nickel oxide (LiNiO$_2$) is an archetypal layered nickelate, but the electronic structure of this correlated material is not yet fully understood. Here we investigate the temperature-dependent speciation and spin dynamics of Ni ions in LiNiO$_2$. Ab initio simulations predict that Ni ions disproportionate into three states, which dynamically interconvert and whose populations vary with temperature. These predictions are verified using x-ray absorption spectroscopy, x-ray magnetic circular dichroism, and resonant inelastic x-ray scattering at the Ni L$_{3,2}$-edge. Charge-transfer multiplet calculations consistent with disproportionation reproduce all experimental features. Our results support a model of dynamic disproportionation that explains diverse physical observations of LiNiO$_2$, including magnetometry, thermally activated electronic conduction, diffractometry, core-level spectroscopies, and the stability of ubiquitous antisite defects. This unified understanding of the material properties of LiNiO$_2$ is important for applications of nickelate materials as battery cathodes, catalysts, and superconductors.

The broad relevance of nickel-based oxides to applications such as energy storage[1], catalysis[2], and superconductivity[3,4], and the possibility to tune their properties by redox and intercalation[5] motivates a rigorous understanding of the rich underlying physics of these materials[6]. Lithium nickel oxide, LiNiO$_2$, is a widely studied model layered nickelate. In catalysis, LiNiO$_2$ has found use as an effective oxygen evolution catalyst[7]. In Li-ion battery cathodes, the formal Ni$^{3+/4+}$ redox couple offers the highest conventional redox capacity for a given cutoff voltage[1]. Despite this broad interest in LiNiO$_2$, however, to our knowledge, no single

model for the electronic structure of LiNiO$_2$ exists that is consistent with all its observed properties.

Since LiNiO$_2$ has previously been comprehensively reviewed in the context of Li-ion batteries[8], here we provide a summary of its key behaviors, including, where relevant, comparisons to other layered alkali metal nickelates $A$NiO$_2$ and rare-earth perovskite nickelates $R$NiO$_3$. The formally 3d$^7$ low-spin ($S = \frac{1}{2}$) configuration of Ni in NiO$_6$ octahedra is orbitally degenerate. Two possible mechanisms for relieving this orbital degeneracy (Fig. 1a) are a symmetry-lowering Jahn-Teller distortion or disproportionation[9,10], whereby different Ni

[1]Dept. of Materials, University of Oxford, Oxford, UK. [2]The Faraday Institution, Harwell Science and Innovation Campus, Didcot, UK. [3]Dept. of Physics and Engineering Physics, University of Saskatchewan, Saskatoon, SK, Canada. [4]Stewart Blusson Quantum Matter Institute, Univ. of British Columbia, Vancouver, BC, Canada. [5]Diamond Light Source, Harwell Science and Innovation Campus, Didcot, UK. [6]Canadian Light Source, Saskatoon, SK, Canada. [7]Dept. of Chemistry, University of Bath, Bath, UK. [8]Present address: Advanced Research Center for Nanolithography, Amsterdam, The Netherlands. ✉e-mail: andrey.poletayev@gmail.com; robert.weatherup@materials.ox.ac.uk; saiful.islam@materials.ox.ac.uk

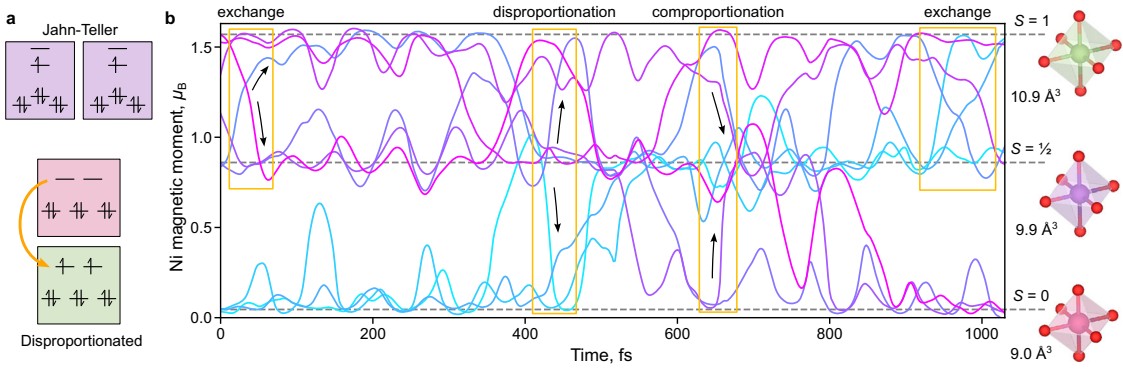

**Fig. 1 | Ab initio simulation of spin dynamics in LiNiO₂. a** Simplified schematic of two pathways of relieving orbital degeneracy: Jahn-Teller distortions preserving spin-half electronic structure (purple), and disproportionation (formal electron donation from pink to green). **b** Ab initio molecular dynamics trajectories of Ni spins in a layer containing nine $NiO_6$ octahedra over 1 ps at 300 K, colored by the initial Ni spin from low (light blue) to high (pink). Exchange, disproportionation, and comproportionation events are highlighted near 50 fs, 420 fs, and 650 fs. $NiO_6$ volumes are annotated for Ni states (green, purple, and pink octahedra).

ions adopt distinct electronic and geometric local environments. Here we define disproportionation simply as the presence of distinct Ni environments and a process of interconversion between them. Considering other layered nickelates, $NaNiO_2$ exhibits a cooperative and collinear Jahn-Teller distortion[11,12], while $AgNiO_2$ exhibits static disproportionation to multiple distinct nickel environments[13–15]. $RNiO_3$ perovskites show similar disproportionation at temperatures below the metal-to-insulator transition, with the oxygens shared unequally between neighboring Ni ions[16–18].

In the case of $LiNiO_2$, both a dynamic non-cooperative Jahn-Teller effect[8,19,20] and a disproportionation of Ni−O bond lengths[21,22] have been proposed, but neither model alone accounts for all the above observations. Here we revisit the mechanism for relieving orbital degeneracy in $LiNiO_2$. We focus on five characteristic behaviors influenced by the local Ni chemistry of $LiNiO_2$:

1. Antisite defects, $Ni_{Li}$, where excess Ni occupies Li sites, are near-impossible to eliminate from $LiNiO_2$, distinguishing it from other layered oxide cathodes[8] and from the sodium analog $NaNiO_2$.
2. $LiNiO_2$ exhibits temperature-activated p-type electronic conductivity[23]. This temperature dependence indicates either Anderson localization or a small-polaron−hopping energy that decreases upon cooling. $LiNiO_2$ with $[Ni_{Li}]$ <3% appears approximately two orders of magnitude more electrically conductive at room temperature than $NaNiO_2$[24], whereas all known polymorphs of $AgNiO_2$ are metallic[25,26].
3. Extended X-ray fine structure (EXAFS) measurements at the Ni K-edge are consistent with distortions of $NiO_6$ octahedra[7,27,28]. These previous studies differ in the direction of the Jahn-Teller distortions assumed when modelling these spectra, and do not consider possible dynamics. Temperature-resolved neutron pair distribution function (PDF) analysis[29] and x-ray diffraction[20] show a gradual transition between cryogenic and room-temperature structures upon heating, rather than an abrupt change of phase.
4. Room-temperature Ni $L_{3,2}$-edge x-ray absorption spectroscopy[22] (XAS) and low-temperature neutron PDF data[29] show substantial differences between $LiNiO_2$ and $NaNiO_2$.
5. The Ni magnetic moments in $LiNiO_2$ are approximately 10% too high for a spin-half $3d^7$ formal state[30], but regain consistency with a formal $Ni^{3+/4+}$ redox process upon delithiation to 50%, i.e., for $Li_xNiO_2$ when $x \leq 0.5$.

Using a combination of ab initio molecular dynamics, three Ni L-edge spectroscopies, and ligand-field multiplet modelling, we show that a dynamic disproportionation model accounts for the five sets of observations above.

## Results

### Dynamic disproportionation

We first examine the behavior of Ni environments in $LiNiO_2$ using spin-polarized ab initio molecular dynamics simulations (Methods). At 300 K (Fig. 1), the spins of Ni ions are principally distributed across three states: below 0.1 $\mu_B$ ($S = 0$), near 0.86 $\mu_B$ ($S = \frac{1}{2}$), or near 1.57 $\mu_B$ ($S = 1$). The spins rapidly convert between these three states via three processes: (i) disproportionation of $S = \frac{1}{2}$ Ni ions to $S = 1$ and $S = 0$, e.g., near 420 fs in Fig. 1b, (ii) the reverse comproportionation, e.g., near 650 fs, and (iii) exchange, e.g., near 50 fs and 900 fs. All three processes preserve an average formal spin-half state of the Ni ions.

The limiting case for this three-state system is a structure consisting of three sublattices in the $NiO_2$ layer, each occupied by Ni exclusively in one of the three spin states[21]. In this limiting case, all $NiO_6$ octahedra are somewhat distorted, with the $S = \frac{1}{2}$ octahedra showing the strongest Jahn-Teller elongation, as expected. In the three-sublattice structure, all bond distances are below 2.10 Å, consistent with EXAFS[7,27,28]. A small departure from hexagonal lattice symmetry (below 1°) is further consistent with neutron scattering and core-level spectra[21,22,29]. We note the similarity between this limiting structure and the three transition-metal sublattices in $Li(NiMnCo)O_2$[31], noncollinear spin models for hexagonal lattices[32], and the disproportionated structure of $AgNiO_2$[13–15].

We next evaluate the ab initio thermodynamics of spin interconversion and disproportionation in $LiNiO_2$. We construct free energy (F) surfaces as $F(s) = - k_BT \ln(p(s))$, where $p(s)$ is the probability distribution of coordinates, $s$, sampled over ab initio trajectories (over 10 ps, Supplementary Information), and $k_B$ and $T$ are Boltzmann's constant and temperature, respectively. As coordinates, $s$, we use Ni magnetic moments and $NiO_6$ volumes, which vary by about 10% with spin states (Fig. 1b). The three Ni states appear as basins in the resulting two-dimensional free-energy surface (Fig. 2a). The magnetic coordinate distinguishes these states more clearly than the $NiO_6$ volume or bond lengths (Supplementary Information), consistent with experiments on perovskite nickelates that demonstrate the primacy of the electronic coordinate[33].

To assess how changes in temperature affect the Ni spin populations, we performed ab initio molecular dynamics at temperatures from 100 K to 600 K. The ab initio free energy surfaces projected onto the spin coordinate (Fig. 2b) show that the spin-zero and spin-one states rise in energy from 100 K to 600 K; hence, disproportionation becomes less favorable with heating. Because the local geometric and electronic coordinates are coupled, changes in the relative populations of the three states provide a possible explanation for the experimentally observed gradual evolution of lattice angle with

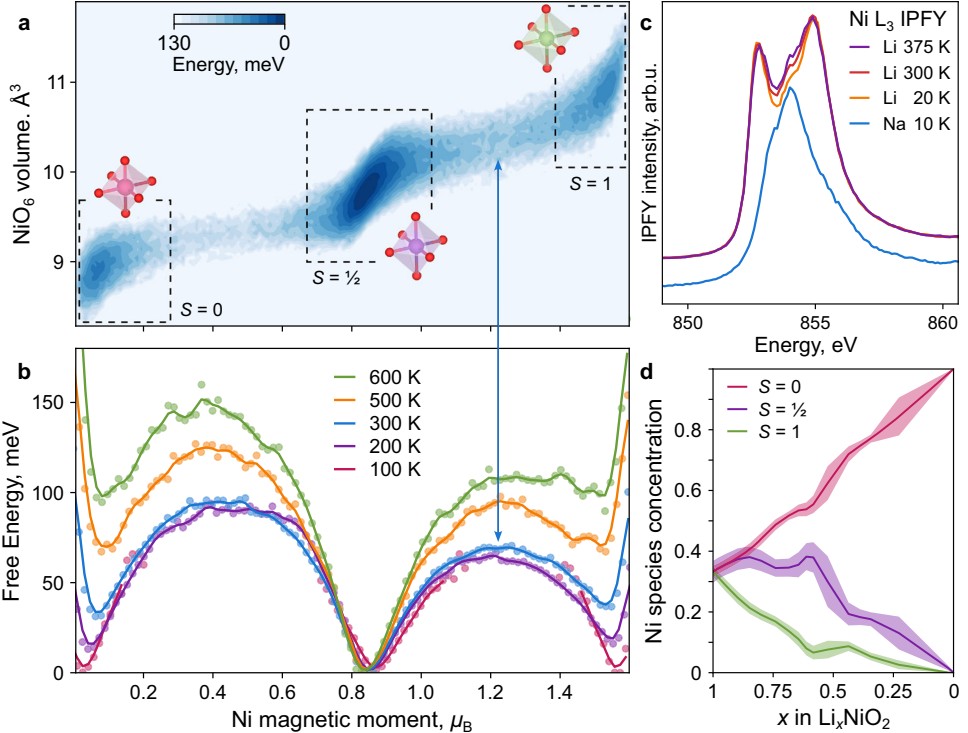

**Fig. 2 | Temperature dependence of spin disproportionation from simulation and experiment. a** Simulated free energy surface at 300 K, versus Ni magnetic moments and NiO$_6$ octahedral volume, with three basins corresponding to spin states highlighted. **b** Simulated free energy profiles versus Ni magnetic moments and temperature. Lines are drawn via a Gaussian filter with bandwidth equal to 1 bin of the histogram that defines the free-energy surface. The arrow connecting (**a**) and (**b**) highlights the saddle point between the $S = \frac{1}{2}$ and $S = 1$ states. **c** Ni L$_3$-edge x-ray absorption spectra of LiNiO$_2$ in inverse partial fluorescence yield (IPFY) mode as a function of temperature. The NaNiO$_2$ spectrum (blue) is offset for clarity. Fits of these spectra to three species are plotted in Supplementary Fig. 1. **d** Concentrations of Ni species (green: $S = 1$, purple: $S = \frac{1}{2}$, pink: $S = 0$) during delithiation from Monte-Carlo sampling of a DFT-based cluster expansion (Methods). The shaded uncertainty values are ±1 s.e. over eight distinct supercell sizes.

temperature[20,29]. At elevated temperatures, the spin-half state predominates, while the overall rate of transitions between states increases (Supplementary Information).

## Spectroscopic verification

We focus on the qualitative temperature trend for experimental validation. The computational prediction of an increasing fraction of $S = \frac{1}{2}$ Ni species with heating is verifiable if spin states can be distinguished experimentally. Core-level spectra are sensitive to changes in the local electronic states, and we, therefore, measured the temperature evolution of the Ni L$_3$-edge XAS in inverse partial fluorescence yield (IPFY) mode[22,34] (Fig. 2c). Two dominant peaks are apparent. Upon heating, these peaks decrease in intensity, while the intensity at the energies between them increases. We therefore discuss these three features in order of increasing energy. First, the low-energy peak is characteristic of NiO-like formally 3d$^8$ species ($S = 1$). Second, the interpeak energy region that grows in intensity with temperature is at an energy that matches the only peak in the corresponding spectrum of NaNiO$_2$ (Fig. 2c, blue). Since NaNiO$_2$ exhibits exclusively a collective Jahn-Teller distortion of $S = \frac{1}{2}$ Ni species (Fig. 1a), we ascribe this middle energy region to $S = \frac{1}{2}$ Ni species in LiNiO$_2$. Third, the high-energy peak could plausibly arise from a lower-spin state such as $S = 0$.

This evolution of the Ni L-edge is analogous to that observed in rare-earth perovskite nickelates $R$NiO$_3$, where double-peaked edge shapes morph into a broad and flat edge with heating across the metal-to-insulator transition[17,18]. The overall temperature evolution of LiNiO$_2$ XAS spectra is weaker than predicted by the increase in the relative proportion of $S = \frac{1}{2}$ Ni species with temperature in our ab initio simulations (Fig. 2b), but the two are qualitatively consistent. We

conclude that LiNiO$_2$ exhibits Ni-disproportionation that is both dynamic and temperature-dependent. Notably, if a Jahn-Teller distortion, collective or not, exclusively accounted for the low-temperature local geometry of LiNiO$_2$, or if disproportionation were only activated with heating, then a stronger semblance to the NaNiO$_2$ spectra would be expected at low temperature, and the evolution of the spectra should be reversed, i.e., the low- and high-energy peaks would be expected to grow with heating.

The continuous rather than abrupt evolution of the L$_3$-edge spectra of LiNiO$_2$ (Fig. 2c) suggests that the mechanism underpinning it differs from that in perovskites, whose spectra switch at the metal-insulator transition: the switching in LiNiO$_2$ is not collective. The continuous evolution of spectra is consistent with an incremental re-equilibration of the fractions of its constituent species at every temperature. Such re-equilibration requires continuous dynamic interconversion and confirms our computational predictions.

Having experimentally validated our model of three-fold dynamic disproportionation, we use this model to predict Ni speciation upon delithiation, as occurs during battery cycling. Using grand canonical Monte-Carlo simulations (Fig. 2d), we predict that during the first half of delithiation (Li content $x > 0.5$ in Li$_x$NiO$_2$), the high-spin Ni species are first to be oxidised, corresponding to net formal Ni$^{2+/4+}$ redox. For $x < 0.5$, the expected Ni$^{3+/4+}$ redox dominates, as reported from bulk-sensitive x-ray Raman scattering[35]. This predicted sequence of redox events is also consistent with magnetometry[30].

## Spectral shapes of nickel species

To understand the origin of the observed changes in spectral features, we perform ligand-field charge-transfer multiplet simulations[36]. Accounting for unequal Ni−O bond lengths arising from both the NiO$_6$

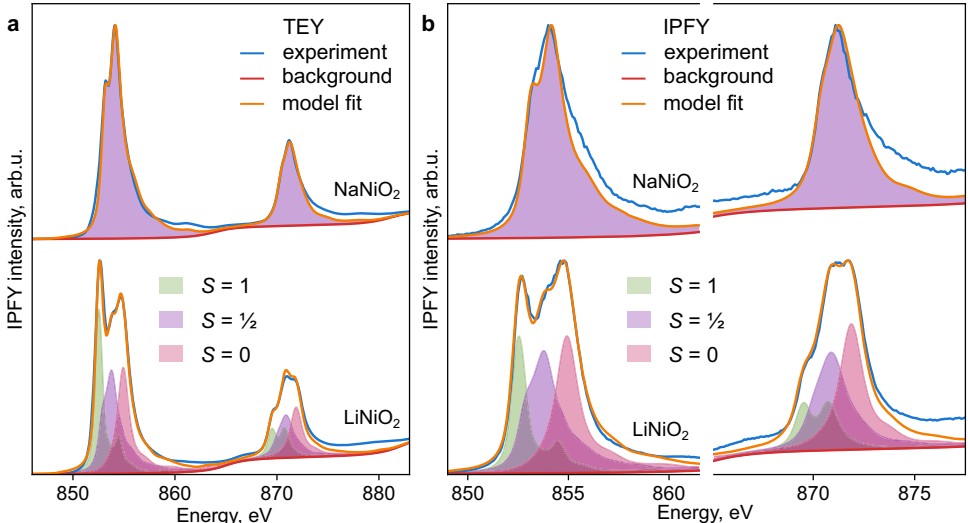

**Fig. 3 | Decomposition of Ni $L_{3,2}$-edge spectra of NaNiO$_2$ and LiNiO$_2$. a** TEY spectra, **b** IPFY spectra. NaNiO$_2$ (top) was measured at 10 K modelled exclusively using the spin-half component (Methods). LiNiO$_2$ (bottom) was measured at 6 K and fit to 42%-35%-23% $S = 1$, $S = \frac{1}{2}$, and $S = 0$ components, respectively, (TEY) or 33%-39%-28% of the same (IPFY). The IPFY $L_2$-edge was rescaled due to saturation.

volume differences and the Jahn-Teller distortion predicted from simulation (Methods) affords a first-principles prediction of state-specific spectral shapes for LiNiO$_2$ and NaNiO$_2$ (Fig. 3 and Supplementary Fig. 1). Our predicted spectra reproduce the experimentally observed TEY and IPFY spectra for both materials. In LiNiO$_2$, the $S = 1$ and $S = 0$ components account for the low- and high-energy $L_3$-edge peaks, respectively. This picture is consistent with both a partial disproportionation and with the usual small Ni excess in LiNiO$_2$, which contributes to the $S = 1$ feature (3–5% in IPFY; Fig. 3b and Supplementary Fig. 1). For the spectra in Fig. 2c, the proportion of $S = \frac{1}{2}$ species grows from 35% at 20 K to 41% at 375 K (Supplementary Fig. 1). Even though a precise quantitative agreement may be beyond the accuracy of the predictions of density-functional theory (DFT), our experimental results are consistent with disproportionation in LiNiO$_2$ and confirm the increase in the proportion of $S = \frac{1}{2}$ ions with temperature. We discuss the sensitivity of computational predictions further in the Supplementary Information.

The calculated partial densities of states for the three Ni species (Supplementary Fig. 2b) verify that both $S = 1$ and $S = \frac{1}{2}$, but not $S = 0$, species contribute to the valence band edge. As with other high-valence Ni compounds[37], strong covalency is predicted here for the $S = \frac{1}{2}$ (mostly $d^8\underline{L}$) and $S = 0$ (mostly $d^7\underline{L}$ and $d^8\underline{L}^2$) species (Supplementary Fig. 2a). A key novelty of our work is the confirmation that these formally high-valence species are present in the pristine, fully lithiated material. Therefore, we next verify the detection of $S = 1$ and $S = 0$ species with x-ray magnetic circular dichroism (XMCD) and resonant inelastic x-ray scattering (RIXS), respectively.

XMCD was performed at the Ni $L_{3,2}$-edge under 8 T applied field (Fig. 4). Circular dichroism is specifically sensitive to unpaired electrons at the Ni centers and can elucidate the competing degrees of charge transfer and covalency[38]. The XMCD spectra thereby assist in constraining the charge transfer multiplet calculations[36]. The $L_3$ XMCD spectra differ between LiNiO$_2$ and NaNiO$_2$ (Fig. 4), mirroring the different x-ray absorption spectra, above. The LiNiO$_2$ $L_3$ XMCD spectrum has a maximum at about 1 eV lower energy and exhibits a sign change near 855 eV in IPFY. The disproportionation model reproduces the XMCD spectra of both compounds in TEY and IPFY modes. The broader dichroism features of NaNiO$_2$ versus the $S = \frac{1}{2}$ Ni species in LiNiO$_2$ are consistent with NaNiO$_2$ exhibiting a stronger Jahn-Teller distortion; XMCD (Fig. 4) appears more sensitive to Jahn-Teller

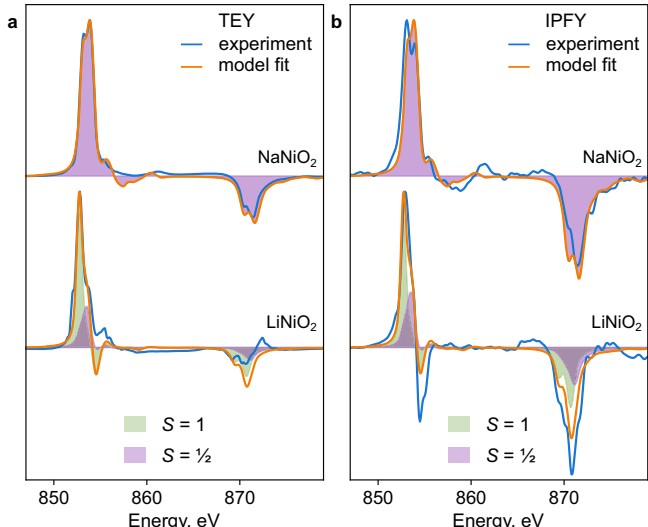

**Fig. 4 | Ni $L_{3,2}$-edge X-ray magnetic circular dichroism (XMCD) of LiNiO$_2$ and NaNiO$_2$. a** TEY, **b** IPFY. NaNiO$_2$ (top) was measured at 10 K and 8 T field, LiNiO$_2$ (bottom) was measured at 10 K and 8 T field. The models (orange) are fit using the same compositions as in Fig. 3. The XMCD signature of the spin-zero component is negligible. The calculated $L_2$ IPFY XMCD was scaled up by the same factor as the linear $L_2$ spectra in Fig. 3b. Raw spectra: Supplementary Fig. 3.

distortions than x-ray absorption (Fig. 3), where the $S = \frac{1}{2}$ shapes are more similar for the two materials. The computed signature of $S = 1$ Ni species in LiNiO$_2$ (green in Fig. 4) includes a sign change characteristic of spinel Ni$^{2+}$, as seen for NiFe$_2$O$_4$ spinel[39,40]. This feature accounts for the lower-energy $L_3$ peak and sign change of the dichroism in LiNiO$_2$ relative to NaNiO$_2$, especially in the more bulk-sensitive IPFY mode. The presence of about 10% excess reduced Ni species near the surface of LiNiO$_2$ observed in TEY mode relative to IPFY (Fig. 3) prevents a more quantitative assignment of the LiNiO$_2$ TEY XMCD spectrum. Nevertheless, the differences between XMCD spectra of the two materials are consistent with the presence of $S = 1$ Ni in bulk LiNiO$_2$ due to disproportionation.

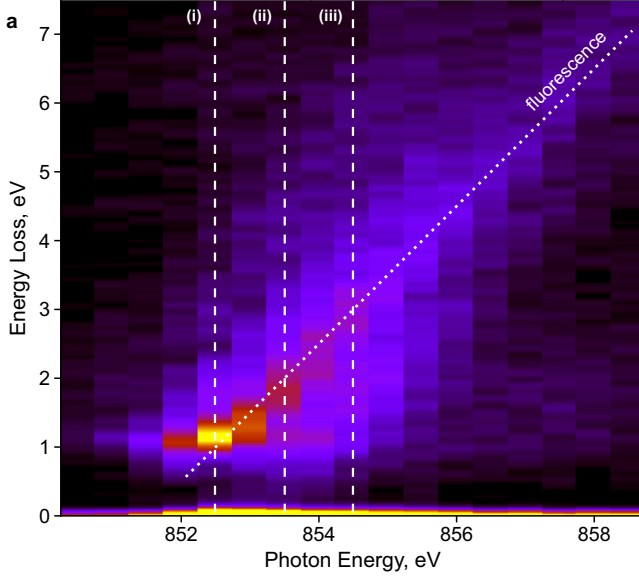

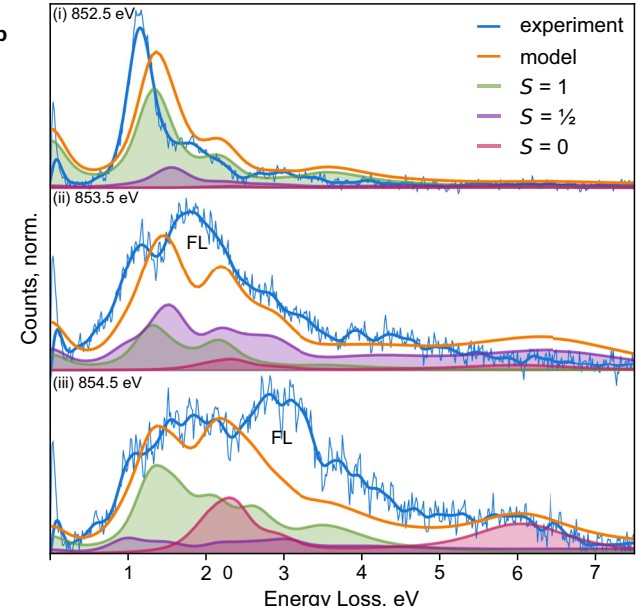

**Fig. 5 | Ni L$_3$-edge resonant inelastic x-ray scattering (RIXS) of LiNiO$_2$. a** RIXS intensity map measured across the L$_3$-edge at 20 K with maximum intensity in yellow and minimum intensity in black, **b** energy loss spectra (blue) at incident photon energies (i) 852.5 eV, (ii) 853.5 eV, and (iv) 855.0 eV compared to calculated loss spectra (orange). Calculated spectra were normalized to 85% of the maximum experimental intensity to account for the fluorescence feature (FL). Relative compositions of nickel species (green, purple, pink) were the same as for IPFY (Fig. 3b). Full calculated d-d and charge-transfer intensity maps are shown in Supplementary Fig. 7.

The $S = 0$ species does not possess an XMCD signature. We, therefore, verify its presence using the added dimension of inelastic energy loss in RIXS. The L$_3$ RIXS map of LiNiO$_2$ (Fig. 5a) includes two features that distinguish it from prior reports of nickelate RIXS[18]. First, there is intensity approaching the elastic line near 853.5 eV, which is about 1 eV higher than in metallic NdNiO$_3$[18]. Second, features near 2 eV and 6 eV loss at 854 eV–855 eV have not, to our knowledge, previously been reported. The fluorescence feature (dotted diagonal in Fig. 5a) extends to <1 eV loss at 852.0 eV, suggesting that LiNiO$_2$ possesses a nonzero optical bandgap[18]. To interpret the RIXS maps, we extended the same Anderson impurity model of three-fold disproportionation as

used to interpret the XAS and XMCD data, without any additional optimization (Supplementary Information), and computed RIXS maps for three Ni species, weighted as for IPFY (Fig. 3b). We discuss loss spectra at three incident photon energies, denoted (i)–(iii) in Fig. 5a.

At 852.5 eV (Fig. 5b, (i)), the main contributions come from $S = 1$ Ni species. This energy loss spectrum is similar to spectra of materials containing d$^8$ states, such as the binary oxide NiO[41] and perovskite NdNiO$_3$[18]. Here, the model slightly over-estimates the crystal field splitting and reproduces the experimental spectrum with a slight shift to higher loss energies. However, surface reduction and the overlap of the main $d$–$d$ excitation with fluorescence near 1 eV loss may contribute to the mismatch here.

At 853.5 eV (Fig. 5b (ii)), low-loss excitations attributed to $S = 1$ and $S = \frac{1}{2}$ Ni species extend to the elastic line, consistent with the presence of states just below and above the Fermi level attributable to both species in DFT calculations (Supplementary Fig. 2b). Broad states above 4 eV loss, above the fluorescence feature (seen at 2 eV loss for this photon energy) and attributable to $S = \frac{1}{2}$ Ni species, likely arise from charge-transfer excitations, consistent with the d$^8$L contribution to its ground state.

Finally, at 854.5 eV (Fig. 5b, (iii)) our model attributes the feature at 6 eV loss exclusively to $S = 0$ Ni species. The high energy loss of this component suggests it is also of charge-transfer origin, but this feature is not present in NdNiO$_3$[18]. Strong contributions of the $S = 0$ species are also evident at 2 eV loss, similar to oxidized Ni species in charged Ni-Mn spinel cathodes[42]. As at lower photon energies, the model slightly overestimates energy loss but reproduces the major spectral features. We conclude that RIXS specifically detects the presence of $S = 0$ Ni species and confirms disproportionation in LiNiO$_2$. Additional weak transitions at 1-2 eV loss above 856 eV (Fig. 5a) are also attributable to the $S = 0$ Ni species (Supplementary Information). While additional fine-tuning of the charge-transfer multiplet model parameters is possible based on the RIXS spectra, we forego this here because of the contributions of reduced surface layers, which likely resemble NiO, to the main $S = 1$ feature, as in TEY (Fig. 3a).

## Consistency with observables
The model of dynamic and temperature-dependent disproportionation presented here is consistent with the five observed behaviors of LiNiO$_2$ detailed above, summarized in order:

1. Additional $S = 1$ Ni ions, arising from disproportionation with the NiO$_2$ layers, are predicted to stabilize Ni$_{Li}$ defects through favorable antiferromagnetic (AFM) interactions (Supplementary Information), explaining the ubiquity of this antisite defect.

2. Activated electronic conduction plausibly arises from exchange (Fig. 1a) between $S = \frac{1}{2}$ and $S = 1$ Ni species. Indeed, the simulated free energy at the saddle point (Fig. 2b) is close to half of the activation energy of electronic conductivity[23]. The increase in this saddle-point energy as the $S = \frac{1}{2}$ state predominates at high temperatures (Fig. 2b) is consistent with increased activation of conductivity upon heating. Electron and hole polarons can be localized in the disproportionated structures, supporting a correspondence between formal spin and charge states (Supplementary Information). In contrast, in NaNiO$_2$, the collective Jahn-Teller distortion precludes the exchange of spin states and reduces electronic conductivity.

3. The small distortion of the LiNiO$_2$ unit cell is consistent with that of the limiting three-fold disproportionated cell, while the gradual decrease in this unit cell distortion with heating[20,29] is consistent with the gradually increasing proportion of $S = \frac{1}{2}$ species.

4. Previously not reported XMCD (Fig. 4), Ni L$_3$-edge RIXS (Fig. 5), and temperature-resolved XAS (Fig. 2c) data provide strong experimental evidence for the disproportionation of Ni species in LiNiO$_2$. Accounting for $S = 0$ and $S = 1$ species affords an interpretation consistent across the Ni L-edge spectroscopies of

LiNiO$_2$ (Figs. 3–5) and of the spectroscopic differences between LiNiO$_2$ and NaNiO$_2$. These observations, combined with charge-transfer multiplet modelling, confirm a negative charge-transfer regime for both compounds[43], but highlight their distinct mechanisms of relieving degeneracy (Fig. 1a). The continuous evolution of L-edge spectra with temperature (Fig. 2c) supports the same pattern: the lack of a phase transition signifies continuous dynamic interconversion between Ni species.

5. The presence of $S = 1$ Ni species in bulk LiNiO$_2$ until 50% delithiation accounts for the increased Ni magnetic moments relative to those expected from formal Ni$^{3+}$ in LiNiO$_2$[30] and is further consistent with bulk-sensitive x-ray Raman scattering[35].

## Discussion

We have identified temperature-dependent dynamic disproportionation as the mechanism relieving orbital degeneracy in the archetypal layered nickelate LiNiO$_2$. Our results support a unified model where Ni species in LiNiO$_2$ exhibit three states with formal spins $S = 0$, $S = \frac{1}{2}$, and $S = 1$ (which correspond to the formal oxidation states Ni$^{4+}$, Ni$^{3+}$, and Ni$^{2+}$, respectively) and interconvert between these on a picosecond timescale. We have verified this behavior with characterization of the nickel L-edge using XAS, XMCD, and RIXS. The low-spin species exhibit strong Ni-O covalency within a charge-transfer multiplet model. These results enable the fingerprinting of the nickel L$_{3,2}$ absorption edges based on first-principles calculations. The temperature dependence and dynamic nature of the disproportionation extend earlier models[21,22] and allow for consistency with a diverse set of experimental observables: thermally activated electronic conductivity[23], local structure from neutron diffraction[29], magnetometry[30], stabilization of antisite Ni excess defects, and, more generally, the gradual changes in the properties of LiNiO$_2$ with heating and delithiation. The fast-timescale dynamic interconversion could be further probed more directly by ultrafast methods such as x-ray photon correlation spectroscopy. Overall, our unified picture of Ni behavior will advance characterisation and understanding of the physics of nickelate materials for a range of applications, including rechargeable batteries, catalysis, and superconductivity.

## Methods

### Sample preparation

Uncoated, polycrystalline LiNiO$_2$ powder was obtained from BASF. NaNiO$_2$ was prepared in house by a solid-state reaction. Appropriate molar amounts of Na$_2$CO$_3$ and NiO were ground together in a pestle and mortar, pressed into a pellet, and then heated at 650 °C under flowing O$_2$ for 12 hours. The heating and cooling rates were controlled at 10 °C min$^{-1}$. Powder X-ray diffraction data were collected for LiNiO$_2$ and NaNiO$_2$ on Cu-source Rigaku diffractometers. GSAS-II software was used to perform the Rietveld refinement analysis. To prepare free-standing electrodes for spectroscopic measurements, cathode powders were mixed with acetylene black and polytetrafluoroethylene (PTFE) as binder in weight ratios 80:10:10, and calendared.

### IPFY XAS and XMCD

Temperature dependent XAS measurements of LiNiO$_2$ were performed at the REIXS beamline of the Canadian Light Source (CLS). Samples were transported to the facility in sealed vials under argon atmosphere, pressed onto carbon tape on copper sample plates under argon atmosphere in a glovebox, and loaded into the x-ray experimental chamber without exposure to atmosphere. Measurements were performed at 20-375 K at pressures below 10$^{-9}$ mBar. The incident beam was horizontally polarized and the normal of the sample plate was aligned with the beam. XAS was collected with TEY by monitoring sample drain current, and IPFY and PFY using a silicon drift detector with ~70 eV resolution. The silicon drift detector was positioned at an angle approximately 60 degrees from the sample normal.

Temperature dependent XMCD and XMLD measurements of LiNiO$_2$ and NaNiO$_2$ were performed in IPFY mode, with simultaneous TEY and FY detection at both the O K and Ni L$_{3,2}$-edges on the high-field magnet end station at the I10 beamline, Diamond Light Source, UK. Powder and electrode samples were mounted onto a copper sample plate using carbon tape in an inert Ar-filled glovebox atmosphere, before being transported directly to the chamber in an Ar-filled sealed transfer vessel (avoiding exposure to air). Measurements were performed at 6-300 K under ultra-high vacuum conditions. The incident beam was directed at a 60° angle to the normal of the sample plate. FY was acquired in the same 60° back-scattering geometry using a Si diode with an Al cover to filter out emitted electrons, mounted in front of the beam entrance port. IPFY was recorded with a four-element Vortex Si drift detector mounted at 90° to the incoming beam (30° to sample normal). XMCD and XMLD measurements were performed at 8 T and collected through the individual detection of right ($\sigma_r$) and left ($\sigma_l$) circular polarizations, or linear horizontal ($\sigma_h$) and vertical ($\sigma_v$) polarizations. The powdered form of the samples means we expect measured signals to be anisotopically averaged, i.e., significant orientation effects are not expected, although this likely reduces the observed extent of dichroism. Both O K-edges and Ni L$_{3,2}$-edges were measured in the continuous scanning mode of the monochromator, with an energy step size of 0.1 eV. All data was divided by the I$_0$ signal to remove top-up intensity spikes and energy-dependent intensity variations associated with the beamline. IPFY data was processed by summing the O emission signal over the incident energy range and following the procedure of Achkar et al.[34]. The pre-edge average background was subtracted, and remaining intensity normalized by the post-edge average.

### Ni L$_3$-edge RIXS

Ni L$_3$-edge RIXS spectra were measured at a temperature of 20 K at the I21 beamline, Diamond Light Source[44]. The incident energy range was 849-859 eV in 0.5 eV steps with energy resolution ≤60 meV. Samples were transferred to the spectrometer using a vacuum-transfer suitcase to avoid air exposure and were pumped down to ultra-high vacuum (UHV) and left to fully degas overnight.

### Computational: DFT, ab initio MD, cluster expansion

DFT simulations were carried out using the projector-augmented wave method[45–47] in the VASP package[48,49] using the meta-GGA functionals SCAN[50] and r$^2$SCAN[51] and forgoing empirical parameters such as a Hubbard $U$ correction or the fraction of exact exchange. The revised Vydrov-van Voorhis (rVV10) non-local dispersion correction was applied. As we were not aware of the accurate parameterization of the rVV10 correction for r$^2$SCAN[52] until substantially after running extended ab initio molecular dynamics simulations using the parameterization for SCAN ($b = 15.7$, $c = 0.0063$)[53], and the favorability of disproportionation was sensitive to the functional over the dispersion correction, the molecular dynamics were not re-run. Static calculations were completed with the parameterization for r$^2$SCAN ($b = 11.95$, $c = 0.0063$), 700 eV plane-wave cutoff, and 0.25 Å$^{-1}$ $k$-point spacing. Energies and forces were relaxed to 10$^{-5}$ eV and 10$^{-2}$ eV/Å, respectively, or better. Ab initio molecular dynamics (AIMD) simulations used a Γ-centered 2×2×2 $k$-point mesh, 2 fs time steps, constant-volume (NVT) ensemble, Nosé-Hoover thermostat with a time constant of 40 steps, electronic convergence of 10$^{-4}$ eV, and the preconditioned conjugate gradient algorithm (VASP ALGO = A), unless specified otherwise.

To identify the states of the Ni we use local spin densities, $S$, as calculated in VASP. This descriptor gives a relatively unambiguous assignment for each Ni without estimating formal charges from the full electronic density in post-processing. The first picosecond of every AIMD run was excluded from analyses for thermostat equilibration. The simulations at 100 K and 200 K, where sampling transitions between Ni states required long trajectories, were initialized by

cooling from 300 K over 500 fs or longer. AIMD simulations with a $Ni_{Li}$ defect were initialized with the starting spin of the antisite Ni set to $-2$ $\mu_B$, and all others as default (1 $\mu_B$). The trajectories of the nickel spins were binned into $S = 0$, $S = \frac{1}{2}$, and $S = 1$ states by milestoning[54] with cutoffs of 0.2 $\mu_B$, 0.7 $\mu_B$, 1.02 $\mu_B$, and 1.4 $\mu_B$. A control simulation in the isobaric (NPT) ensemble was carried out with the Langevin thermostat coupled only to the Li atoms at 12 ps$^{-1}$ to avoid perturbing the dynamics of Ni-O bonding.

A decorated cluster expansion of defect-free $LiNiO_2$ was trained to predict the nickel speciation on delithiation[55]. Reference structures for training were chosen to be large enough to allow for disproportionation should that be favorable (4-12 Ni ions per layer, 48-144 atoms), and pre-distorted for accelerating relaxation. The DFT settings for reference structures were as for static calculations above, although some relaxations were shortened when clearly approaching convergence due to the reduced requirements on precision for the purposes of the cluster expansion. The root mean squared errors (RMSE) were 4.6 meV/f.u. over the training set and 5.6 meV/f.u. over the hold-out validation set. Charge-neutral grand canonical Monte-Carlo (CNGCMC) sampling[56] was used to estimate the nickel speciation at all states of delithiation (Fig. 2d), with spin states used as formal charge states for nickel. To mitigate the effects of commensurate lattice orderings[57] on predicted speciation, eight different supercell sizes were averaged. For each chemical potential of Li vacancies, the CNGCMC runs were initialized at 1000 K, cooled to 100 K for finding the ground state, heated to 500 K, and sampled for $10^6$ steps, with the first half of those discarded. The concentrations of Ni species were averaged over supercells for each chemical potential of Li vacancies[58]; chemical potentials of Ni species were kept at zero relative to each other. A more detailed study of delithiation in $LiNiO_2$ and the limitations of the cluster expansion formalism is the subject of follow-on work.

Defect formation energies were calculated only for charge-neutral structures from relaxed defect-free and defect-incorporating cells[59-62]. The chemical potentials of the elements at synthesis conditions were calculated from the energies of the reference phases[62-64]. At the typical conditions of synthesis—1 atm $O_2$ pressure and 700 °C—the chemical potential of oxygen is $\mu_O = -1.065$ eV, which determines $\mu_{Li} = -2.962$ eV and $\mu_{Ni} = -1.379$ eV. We account for the antiferromagnetic–paramagnetic transition of NiO at its Néel temperature by taking the energy of paramagnetic NiO as the average of computed AFM and FM energies.

### Multiplet ligand field theory modelling of the Ni $L_{3,2}$-edge

The nickel $L_{3,2}$-edge multiplet ligand field theory (MLFT) simulations were performed using the many-body code Quanty[65]. The simulation was implemented using a single-cluster $NiO_6$ Hamiltonian of $O_h$ symmetry for $S = 0,1$ and $D_{4h}$ symmetry for $S = \frac{1}{2}$. The Ni 2p, Ni 3d, and ligand shells are explicitly included. For all calculations, Slater integrals are scaled to 80% and 85% for the initial and final Hamiltonians, respectively. Additionally, onsite ligand energy shifts of $T_{pp} = \pm 0.75$ eV were applied to the ligand orbitals of $e_g$ (+) and $t_{2g}$ (-) symmetry.

A charge transfer energy of $\Delta = -0.5$ eV assumed for the $3d^7$ $S = \frac{1}{2}$ Ni, as used by Green et al.[36]. This charge transfer energy, along with a Coulomb interaction energy of $U_{dd} = 6$ eV, leads to charge transfer energies of 5.5 eV and -6.5 eV for the $S = 1$ ($3d^8$) and $S = 0$ ($3d^6$) clusters, respectively. A core-valence Coulomb interaction parameter of $U_{pd} = 7$ eV was used, which is the standard -1 eV larger than $U_{dd}$. Hopping integrals and crystal field energies are obtained directly from bond lengths using Harrison's formulas[36,66], and hopping integrals were scaled by 80% in the XAS final state[36]. The DFT-determined bond lengths were used for the three sites in $LiNiO_2$. For $NaNiO_2$, bond lengths of 1.93 Å and 2.16 Å for x/y and z bonds were used, respectively[11,12,67], which yields a slightly larger Jahn-Teller distortion

than for the $LiNiO_2$ $S = \frac{1}{2}$ site geometry. To obtain the $d^x L^y$ terms for the ground-state configurations, the wavefunctions are projected onto the corresponding basis set in Quanty. The charge transfer energies, hopping integrals, and crystal field potential energies are listed below for all calculations.

$S = 1$ calculation (eV): $\Delta = 5.5$, crystal field $10D_q = 0.71$, hopping integrals $V_{eg} = 2.63$, $V_{t2g} = 1.52$.

$S = \frac{1}{2}$ calculation (eV): $\Delta = -0.5$, $10D_q = 0.78$ with Jahn-Teller splitting of $\Delta_{eg} = 0.15$ and $\Delta_{t2g} = 0.10$. Here, $\Delta_{eg}$ denotes the difference between the $x^2 - y^2$ and $3z^2 - r^2$ onsite energies, and $\Delta_{t2g}$ the difference between the $xy$ and $xz/yz$ onsite energies (eV): $V_{3z2-r2} = 2.43$, $V_{x2-y2} = 3.33$, $V_{xz/yz} = 1.41$, $V_{xy} = 1.93$.

$S = 0$ calculation (eV): $\Delta = -6.5$, $10D_q = 0.93$, $V_{eg} = 3.456$, $V_{t2g} = 2.004$. $S = \frac{1}{2}$ calculation for $NaNiO_2$ (eV): $\Delta = -0.5$, $10D_q = 0.70$ with Jahn-Teller splitting of $\Delta_{eg} = 0.19$ and $\Delta_{t2g} = 0.12$. $V_{3z2-r2} = 2.02$, $V_{x2-y2} = 3.17$, $V_{xz/yz} = 1.17$, $V_{xy} = 1.84$.

### Data availability

Computed Ni $L_{3,2}$-edge spectral shapes, computed Ni $L_{3,2}$-edge RIXS spectra, experimental IPFY and RIXS spectra, and exemplar ab initio molecular dynamics trajectories with setup files are available at reference[68].

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

## Acknowledgements

The authors acknowledge funding from the UK Faraday Institution (faraday.ac.uk; EP/S003053/1, FIRG016, FIRG024, FIRG030) and the European Research Council (ERC) under the European Union's Horizon 2020 research and innovation programme (EXISTAR, grant agreement No. 950598). B.J.M. acknowledges support from the Royal Society (URF/R/191006). R.S.W. acknowledges a CAMS-UK Fellowship through the Analytical Chemistry Trust Fund and a UKRI Future Leaders Fellowship (MR/V024558/1). R.J.G. and G.H. acknowledge funding from the Natural Sciences and Engineering Research Council of Canada (NSERC). The authors acknowledge the HEC Materials Chemistry Consortium (EP/R029431) for the use of Archer2 high-performance computing (HPC) facilities. The authors also acknowledge the Faraday Institution's Michael HPC resource. We acknowledge Diamond Light Source for time on beamlines I10 and I21 under proposals MM33062 and MM30644-1, and Dr. Stefano Agrestini, Dr. Mirian Garcia-Fernandez, and Dr. Ke-Jin Zhou for assistance with the RIXS measurements. We acknowledge the support of the Royal Academy of Engineering under the Research Fellowship scheme. Part of the research described in this paper was performed at the Canadian Light Source, a national research facility of the University of Saskatchewan, which is supported by the Canada Foundation for Innovation (CFI), the Natural Sciences and Engineering Research Council (NSERC), the Canadian Institutes of Health Research (CIHR), the Government of Saskatchewan, and the University of Saskatchewan. A.D.P. is grateful to Dr. Pezhman Zarabadi-Poor and Dr. Gregory Rees for insightful discussions.

## Author contributions

Initial investigations of $LiNiO_2$ were carried by J.P.C. and extended to AIMD by A.D.P. with advice and supervision from M.S.I. and B.J.M. The temperature-dependent XAS experiments were proposed by A.D.P. and carried out by R.J.G. and R.S. (CLS). The XMCD experiments were proposed by R.J.G., J.E.N.S., R.S.W., and A.D.P., and carried out by A.D.P., J.E.N.S., L.A., and L.J. with P.B. (DLS). The RIXS experiments were proposed by A.D.P. and R.A.H. and carried out by R.A.H. Samples were prepared by R.A.H., L.A., J.E.N.S., and L.J. The charge-transfer multiplet modelling was carried out by G.H. and R.J.G. A.D.P. led the writing of the manuscript with input and contributions from all authors.

## Competing interests

The authors declare no competing interests.
