## [Transparent Peer Review file · Nature Communications]

Temperature-Dependent Dynamic Disproportionation in LiNiO_2

Corresponding Author: Dr Andrey Poletayev

Version 0:

Reviewer comments:

Reviewer #1

(Remarks to the Author)

This manuscript offers a number of experimental and computational results that are meant to support an electronic structure model that coherently explains the behaviors of LiNiO_2 (LNO), i.e. temperature-dependent dynamic disproportionation, in which three distinct states with a formal oxidation Ni^{2+} , Ni^{3+} , and Ni^{4+} fluctuate on a picosecond timescale. Later in the text, the authors claim the confirmation of the presence of formally high-valence species (d_{8L} , d_{7L} and d_{8L2}) in (fully-lithiated) LNO as a key novelty.

While a lot of careful work has been into this manuscript, its novelty aspects as well as the central message being put forward could be strengthened more. For instance, “fingerprinting Ni L spectra” with the above-mentioned oxidation states is rather well-known at least from a spectroscopic viewpoint. Nevertheless, I still am in favor of publishing this work, among other things, due to its potential to raise awareness of the importance of ligand-metal interplay in nickelates, particularly due to anionic participation in explaining important properties. I recommend to address this point as well as the other points outlined below.

The presence of fluctuations on a picosecond timescale between the above-mentioned Ni oxidation states relies much on the authors’ molecular dynamics simulations. However, there doesn’t seem to be a direct experimental method to verify this effect. Instead, the authors compare indirect temperature effects that appear largely consistent with experiments, e.g. referencing lattice angle changes as well as own x-ray absorption spectroscopy. Comparisons to battery cycling are also made. A recent battery EES paper (DOI: 10.1039/d3ee04354a) shows that LNO delithiation leads to a core-shell structure, where the shell contains ions close to Ni^{2+} and the core close to Ni^{4+} . However, O_2 oxygen redox is substantial as well. How does/would the model in the present work incorporate this important observation of anionic activity? How would the authors relate this effect to the presence of d_{8L} , d_{7L} and d_{8L2} configurations that have “ligand holes” rather than just Ni holes?

Also, the authors mention that the temperature effect on the XAS spectra turned out to be stronger than observed whereas the authors claim qualitative agreement. I did not find a direct comparison of the prediction to the experiment or an attempt to a possible explanation to the discrepancy (or its magnitude). Could the authors please elaborate!

The authors’ model does seem to work well in describing the evolution of the Ni redox of LNO (first $\text{Ni}^{2+/4+}$ then $\text{Ni}^{3+/4+}$) based on X-ray Raman scattering and magnetometry. Fig. 2d in the present manuscript is derived to predict the distribution of the three different Ni species at different lithiation stages. In view of attaining a broader context of the involved “Ni-oxide” dynamics it would be valuable that the authors discuss the applicability of their model to related nickelate Li-compounds. For instance, the battery cathode $\text{Li}_{1-x}\text{Ni}_x\text{Mn}_{1.5}\text{O}_4$ (LNMO) appears to show very similar transitions between Ni oxidation states as a function of lithiation degree by studying Ni L XAS (J. Phys. Chem. C 119, 27228, 2015). Regarding the Ni L RIXS spectra of LNO (Fig. 5b), I noticed that the energy loss feature around 2-2.5 eV observed by the present manuscript’s authors, while being absent in NdNiO_3 , may be related to a feature observed for LNMO (actually $\text{Li}_{1-x}\text{Ni}_x\text{Mn}_{1.56}\text{O}_4$) that is attributed to ligand hole states by the authors of Energy Adv. 2, 375 (2023). Is the disproportionation model (in principle) extendable to the above situation? How so or why not?

Reviewer #2

(Remarks to the Author)

Summary: Poletayev et al. uses DFT and DFT-based molecular dynamic simulations to obtain an atomic and electronic structural model for LiNiO₂ at high and low temperatures. This model is not the thermodynamically most stable one (which is a few meV/atoms lower), and is dependent on the calculation parameters (in particular the functional). The obtained model is composed of bond and charge (spin) disproportionated Ni sites with dynamic exchange between Ni sites. The theoretical thermodynamic phase obtained from relaxing the high symmetry rhombohedral phase contains Jahn-Teller distorted Ni sites. Poletayev et al. also report experimental Ni L-edge spectra, namely variable temperature XAS, XMCD and RIXS with variable excitation energy. Single cluster ligand field multiplet calculations based on bond/charge disproportionated model reported by the authors allows a decent reproduction of these Ni L-edge spectroscopies. Finally, Poletayev et al. discuss the reported model with regard to previously reported electronic conductivity and Li/Ni antisite defects in LiNiO₂. LiNiO₂ conductivity is thermally activated and higher than JT distorted NaNiO₂, which the authors attribute to polarons hopping in the bond/charge disproportionated LiNiO₂ system. Ni²⁺ antisite mixing is proposed to be facilitated by the presence of Ni²⁺ in the charge disproportionated LiNiO₂.

General comment: Considering the amount of recent work dedicated to understand LiNiO₂ local and electronic structure, I believe this topic is of interest to the battery/physics community. The novelty of Poletayev et al. work is nested into (1) the dynamic nature of the charge/bond disproportionation, (2) new interesting spectroscopic datasets (variable temperature Ni L-edge XAS, XCMD and variable excitation energy Ni L-edge RIXS), (3) multiplet calculations of XMCD and RIXS and comparison with the experimental data. Note that the static bond/charge disproportionation model have been reported before [10.1103/PhysRevB.100.165104], while the dynamic nature of LiNiO₂ postulated by Sicolo et al. [10.1021/acs.chemmater.0c03442]. In terms of datasets, some XAS and RIXS were reported by Jacquet et al. [10.1002/aenm.202401413]. I believe that the modelling, spectroscopic simulations, data acquisition and analysis is well done and generally close to state-of-the-art.

Questions:

- (1) Do I understand correctly that the disproportionated model is obtained by relaxing a starting structural model which is already disproportionated? As suggested by "When this asymptotic three fold disproportionated structure is relaxed ..." in the supplementary information? Along the same line, the paragraph starting by "The limiting case of the three-state system" is unclear because it mixes theoretical results and literature. Clarifying how the model is obtained in the main text would be welcome (even if it is not the most stable system, as the authors mention theoretical calculations on LiNiO₂ a correlated, locally distorted system potentially dynamic might be a the limit of current theories).
- (2) The variable temperature Ni L-edge IPFY is interesting (especially the use of IPFY which is currently the best way to measure intensities in fluorescence mode, to the best of my understanding). However, the changes with temperature are really small. $S = \frac{1}{2}$ specie phase fraction only changes from 35% to 37% heating from 25 K to 300 K. That's in the experimental error if we consider that the reproducibility of the phase fraction measurement is 2% (the reproducibility between two measurements as mentioned by the authors in the figure caption of extended figure 1). Can the author provide a more accurate reproducibility? Otherwise, I'm afraid to say that the temperature change in $S = 0$, $S = \frac{1}{2}$ and $S = 1$ population is not experimentally supported.
- (3) The statement "a key novelty of our work is the confirmation that these formally high valence species are present in the pristine, fully lithiated material" is debatable. Indeed, other works have made similar observations recently [10.1021/acsenerylett.4c00360, 10.1002/aenm.202401413]
- (4) In the RIXS interpretation, I find difficult to claim "the fluorescence feature [...] extends to < 1 eV" without performing a decomposition of the RIXS signal into "raman-like" and "fluorescence-like" signal before as Bisogni et al. performed for example [10.1038/ncomms13017]. Currently, this statement is merely based on visual observation.
- (5) Why would the "gradual decrease in this unit cell distortion with heating" be consistent with "gradually increasing proportion of $S = \frac{1}{2}$ species"? NaNiO₂ has plenty of $S = \frac{1}{2}$ species but is distorted.
- (6) How did the authors obtained the fraction of d₇, d_{8L}, d_{9L2} ... configuration shown in extended Figure 2. Also, the bond lengths are mentioned in the figure caption but not shown in the figure.
- (7) typo in the energy resolution of the XAS (70 ev...)
- (8) Electrodes has been measured for temperature dependant XMCD but there formulation/preparation is not described.
- (9) Please include the energy resolution for the RIXS measurement
- (10) Regarding the multiplet calculations, I wonder why the authors didn't use double site multiplet calculations has performed for the nickelates? [10.1103/PhysRevX.8.031014]

Reviewer #3

(Remarks to the Author)

This is a fascinating and impressive piece of work which brings together a range of state-of-the-art computational and experimental techniques to probe the electronic structure of LiNiO₂ - a material of intrinsic interest and importance owing to its applications in battery technology. The present referee has most expertise in the computational techniques employed and can confirm that these are appropriate and have been carefully employed. The results demonstrating dynamic disproportionation are particularly interesting and novel and have wider implications for our understanding of the electronic structure of transition metal oxides. The work is of the quality and originality to justify publication in Nature Comms. I have two recommendations for minor changes:

1. The authors comments that the system has a high degree of covalence. Could they for at least some configurations provide a Bader charge analysis, which would be helpful additional information.

2. Can they clarify whether the DFT calculations were performed on the unit cell or on a supercell.

Version 1:

Reviewer comments:

Reviewer #1

(Remarks to the Author)

The authors have addressed the raised concerns and questions adequately and satisfactorily. I support publication of the paper.

Reviewer #3

(Remarks to the Author)

The authors have responded appropriately to my suggestions and I can now recommend this very interesting and novel work for publication in Nature Comms.

We thank the editor and the reviewers for their time and their consideration of our manuscript. Below, reviewer comments (**R1**, **R2**, **R3**) are in **bold**, and alterations to the manuscript's text are in *italic*.

R1: This manuscript offers a number of experimental and computational results that are meant to support an electronic structure model that coherently explains the behaviors of LiNiO₂ (LNO), i.e. temperature-dependent dynamic disproportionation, in which three distinct states with a formal oxidation Ni²⁺, Ni³⁺, and Ni⁴⁺ fluctuate on a picosecond timescale. Later in the text, the authors claim the confirmation of the presence of formally high-valence species (d⁸L, d⁷L and d⁸L²) in (fully-lithiated) LNO as a key novelty. While a lot of careful work has been into this manuscript, its novelty aspects as well as the central message being put forward could be strengthened more. For instance, “fingerprinting Ni L spectra” with the above-mentioned oxidation states is rather well-known at least from a spectroscopic viewpoint. Nevertheless, I still am in favor of publishing this work, among other things, due to its potential to raise awareness of the importance of ligand-metal interplay in nickelates, particularly due to anionic participation in explaining important properties. I recommend to address this point as well as the other points outlined below.

We thank the reviewer for their time and careful reading of our manuscript. We might add that we believe the larger value to readers and the broader community is the consistency between our model and a large number of previously un- or under-explained behaviours in LiNiO₂. Our model, unlike previous ones, is a unifying one. We also note that we are unaware of any decompositions of multi-species Ni L-edge spectra for layered nickelates from first principles. Our own work (ref. 36 in the main text, now published as [1]) building on the model we put forward in this manuscript further demonstrates that the use of endpoint spectra (e.g. LiNiO₂ and fully delithiated NiO₂) as references is not appropriate.

R1: The presence of fluctuations on a picosecond timescale between the above-mentioned Ni oxidation states relies much on the authors' molecular dynamics simulations. However, there doesn't seem to be a direct experimental method to verify this effect. Instead, the authors compare indirect temperature effects that appear largely consistent with experiments, e.g. referencing lattice angle changes as well as own x-ray absorption spectroscopy.

The reviewer is correct: directly probing picosecond-timescale fluctuations in Ni-O coordination environments would be challenging. We can think of several ways, informed in part by parallels to nickelate superconductors, for example single-pulse split-delay x-ray photon correlation spectroscopy at an x-ray free-electron laser. Alternatively, since the changes in spectral intensity at the L-edge occur slowly over a wide range of temperatures (Figure 2c), a transient temperature jump experiment would have to raise the temperature of the material by several tens of degrees, which is also challenging considering possible sample damage. Sample preparation is further likely to be complex for such studies. We

hope to carry out these experiments in the future, but the reviewer will surely agree that experiments of such complexity would merit their own manuscripts.

The key evidence that our multi-temperature measurement provides is in demonstrating that the changes in L-edge spectra do not arise from an abrupt phase transition, but instead the spectra evolve incrementally with temperature. The only way for the concentrations of three species to re-equilibrate in such a continuous manner is for those species to interconvert all the time. This is further consistent with the smooth evolution of lattice parameters that we note in the introduction (item 3 in our list). This behavior is in contrast, for example, with that of perovskite rare-earth nickelates, which switch abruptly between two states (disproportionated insulator to comproportionated metal) at a well-defined temperature. We recognize that this argument has not been sufficiently detailed in the main text and take this opportunity to improve the manuscript.

We modify the manuscript to accentuate these points:

- *“The continuous rather than abrupt evolution of the L_3 -edge spectra of LiNiO_2 (Figure 2c) suggests that the mechanism underpinning it differs from that in perovskites [references], whose spectra switch at the metal-insulator transition: the switching in LiNiO_2 is not collective. The continuous evolution of spectra is consistent with an incremental re-equilibration of the fractions of its constituent species at every temperature. Such re-equilibration requires continuous dynamic interconversion and confirms our computational predictions.”* (spectroscopic verification)
- *“The continuous evolution of L-edge spectra with temperature supports the same pattern: the lack of a phase transition signifies continuous dynamic interconversion between Ni species.”* (consistency with observables)
- *“The fast-timescale dynamic interconversion could be further probed more directly by ultrafast methods such as x-ray photon correlation spectroscopy.”* (conclusions)

R1: Comparisons to battery cycling are also made. A recent battery EES paper (DOI: 10.1039/d3ee04354a) shows that LNO delithiation leads to a core-shell structure, where the shell contains ions close to Ni^{2+} and the core close to Ni^{4+} . However, O_2 oxygen redox is substantial as well. How does/would the model in the present work incorporate this important observation of anionic activity?

Our work quantifying Ni and O speciation from bulk-sensitive x-ray Raman spectroscopy (XRS, ref. 36, now published as [1]) using the model presented here addresses more directly the subject of forming molecular O_2 . Juelsholt *et al.* [2] claim the formation of about 2% molecular O_2 on the first cycle (their Fig. 3) based on RIXS intensities. This should correspond to Ni reduction, which at the 1-2% level has two possible origins: Ni_{Li} defects, and O_2 formation. Our bulk-sensitive spectra do not rule out either possibility: in ref. 36 we calculate about 5% Ni^{2+} at 4.8 V.

However, more substantial transition metal reduction occurs near the surface of cathode particles (up to about 200 nm depth) already on the first cycle. This is evident from the

differences between fluorescence-yield (FY) and XRS in ref. 36: FY spectra show significant Ni reduction relative to the XRS. This is confirmed with complementary electron energy loss spectroscopy (EELS) depth profiles. The “shell” containing reduced Ni may be comparable to the probing depth of both FY and RIXS, i.e., thicker than previously considered. We agree the formation of bulk molecular O₂ deserves further study. We recently calculated that the formation of confined O₂ remains thermodynamically favorable in NiO₂, although the driving force is smaller relative to that for evolving O₂ gas [3].

R1: How would the authors relate this effect to the presence of d⁸L, d⁷L and d⁸L² configurations that have “ligand holes” rather than just Ni holes?

Formal oxidation states are the simplest and most useful framework to communicate the basic idea that there are three distinct Ni species in LiNiO₂, one of which is the same as the Ni species in charged NiO₂. The three Ni species do not correspond 1:1 to the d^xL^y multiplet basis set, and as such the formal oxidation states are a more effective basis set.

Our use of formal oxidation states should not be taken to imply complete ionicity or “Ni holes”. We already use the qualifier “formal” at every mention of oxidation states in the main text to avoid such interpretation. However, we see no conflict between using formal oxidation states to denote distinct Ni species and the strong covalency of Ni-O bonding at high states of charge. The d^xL^y contributions for all three Ni species are listed in Extended Data Figure 2. In the present work, we are careful to not over-use the d^xL^y nomenclature since these terms do not map to Ni species and do not offer predictive power without the additional information which we provide. Overall, we do not see utility in emphasizing or discussing the negative charge transfer behavior of NaNiO₂ and LiNiO₂ in the present work: it does not explain the differences between them. We further gave an example in the main text: LiNiO₂ and NaNiO₂ are distinguished not by the overall extent of negative charge transfer, but by the differences in Ni speciation.

As an aside, the magnetic moments, which map well to formal oxidation states, are an especially useful metric for modelling redox behavior [4,5]. The success of the charge-decorated cluster expansion in matching x-ray Raman spectra (compare Figure 2d in the main text with Supplementary Figures 2-5 of ref. 36 and [1]) further validates our approach. The only information shared between the cluster expansion and the fitting of L-edge spectra is the use of local geometries computed at the r²SCAN+rVV10 level of theory as a starting point informing ligand-field multiplet calculations, yet the percentages of Ni species from the two approaches are in good agreement at all states of charge, modulo Ni_{Li} defects. We see no path to this level of predictive power from approaches focusing on the negative charge transfer character of these layered nickelates. However, while the cluster expansion captures the configurational entropy of disproportionation, it is missing the possible vibrational energy and vibrational entropy contributions, which may differ across Ni species and could also contribute to the temperature evolution of Ni speciation.

R1: Also, the authors mention that the temperature effect on the XAS spectra turned out

to be stronger than observed whereas the authors claim qualitative agreement. I did not find a direct comparison of the prediction to the experiment or an attempt to a possible explanation to the discrepancy (or its magnitude). Could the authors please elaborate!

We are happy to clarify. In this study, we have used ab initio modelling to inform spectroscopic experiments (temperature dependence of x-ray absorption, Figure 2c), some of which are rarely if ever conducted in the battery context. The experimentally observed temperature effect on the XAS spectra (Figure 2c) is weaker than predicted from ab initio molecular dynamics (Figure 2b). Using the milestoning approach discussed in the methods, we obtain at 200 K 58% $S = \frac{1}{2}$, at 300 K 67%, at 500 K 72%, and at 600 K 78% from ab initio molecular dynamics. Below 200 K, achieving sufficient sampling is challenging, and we discuss our verification of the sufficiency of simulation lengths in the supplementary information. There can be many possible reasons for the quantitative disagreement with experiment, but ultimately density-functional theory is far from a perfect oracle. Despite very small simulation volumes and sensitivity to the level of theory and reciprocal-space sampling, the simulations have provided valuable information for experimental studies. Given that computational predictions can be even qualitatively sensitive to the level of theory employed (see, e.g., Figures S6-7), we focus on the qualitative temperature trend to avoid over-interpreting computational results.

Chronologically, we first carried out simulations at the SCAN and r²SCAN+rVV10 levels of theory and observed a qualitative difference between them: SCAN predicts disproportionation to be favored with heating, r²SCAN+rVV10 with cooling. This informed our choice of experimental probes to determine which one is qualitatively correct.

We quote from the first pre-print version of this manuscript, posted to arXiv (2211.09047v1) prior to conducting experimental verification: “Computational predictions such as ones put forward here require additional experimental verification. Both the experimental studies highlighted above, and our computational predictions suggest that [LiNiO₂] changes substantially and gradually between cryogenic, ambient, and synthesis temperatures. The sensitivity analysis above assists in finding probative experiments to verify the computational predictions. We expect structural probes (diffraction, PDF) to be most probative towards cryogenic temperatures, as exploited by Foyevtsova et al. [ref]. We expect data from structural probes to become harder to interpret for distinguishing local chemistry at room and elevated temperatures because of the dynamical broadening of bond distances, the multitude of possible bond distances, and the poor separation of the various states along the structural coordinate. Bulk-sensitive core-level spectroscopic probes should retain sensitivity, however, due to the more effective separation of nickel states along the magnetic coordinate and the possibility for distinguishing states along additional coordinates such as energy loss in inelastic X-ray scattering. Therefore, we expect self-consistent spectroscopic data sets spanning cryogenic to ambient temperatures to potentially offer a more complete picture of LiNiO₂.”

The cluster expansion yields, in absolute terms, better agreement with experiment with regard to the percentages of Ni species than the ab initio molecular dynamics. We hope to understand that better by incorporating antisite defects into the cluster expansion in follow-on work.

R1: The authors' model does seem to work well in describing the evolution of the Ni redox of LNO (first $\text{Ni}^{2+/4+}$ then $\text{Ni}^{3+/4+}$) based on X-ray Raman scattering and magnetometry. Fig. 2d in the present manuscript is derived to predict the distribution of the three different Ni species at different lithiation stages. In view of attaining a broader context of the involved "Ni-oxide" dynamics it would be valuable that the authors discuss the applicability of their model to related nickelate Li-compounds. For instance, the battery cathode $\text{Li}_{1-x}\text{Ni}_{0.44}\text{Mn}_{1.56}\text{O}_4$ (LNMO) appears to show very similar transitions between Ni oxidation states as a function of lithiation degree by studying Ni L XAS (J. Phys. Chem. C 119, 27228, 2015). Regarding the Ni L RIXS spectra of LNO (Fig. 5b), I noticed that the energy loss feature around 2-2.5 eV observed by the present manuscript's authors, while being absent in NdNiO_3 , may be related to a feature observed for LNMO (actually $\text{Li}_{1-x}\text{Ni}_{0.44}\text{Mn}_{1.56}\text{O}_4$) that is attributed to ligand hole states by the authors of Energy Adv. 2, 375 (2023). Is the disproportionation model (in principle) extendable to the above situation? How so or why not?

There are indeed substantial similarities between spectral shapes calculated by Qiao et al. (2015) for LNMO and ours, despite differences in the magnitudes of parameters used, e.g., the crystal field energies ($10D_q$). We used Harrison's rules informed by DFT bond distances to calculate hopping integrals and crystal field energies and further used the XMCD spectra to validate the fitting of the XAS and reduce ambiguity arising from broadening effects. An estimate of the errors arising from using DFT geometries, which have slightly shorter Ni-O bond distances than experiment (unit cell a, b 2.85 Å vs 2.88 Å), is the overestimation of charge-transfer energies which leads to an over-estimation of loss in RIXS spectra (Figure 5b), noted in the main text. The transferability of our fitting to other nickelate compositions is the subject of ongoing work, for example with mixed (Ni,Mn,Co) oxide cathodes.

Regarding the RIXS features near 2-2.5 eV at the high excitation energy 854.5 eV (our Figure 5b), we attribute them primarily to the $S = 0$ species as discussed in the main text. This mostly covalent species with over 90% content of $d^{x>6}L^{y>0}$ (Extended Data Figure 2) should indeed be similar to the Ni species in charged LNMO. We now note this point in the main text: "Strong contributions of the $S = 0$ species are also evident at 2 eV loss, *similar to oxidized Ni species in charged Ni-Mn spinel cathodes [ref. Massel et al.]*".

However, our model predicts this species to yield two distinct spectral contributions at 2-2.5 eV loss and 6 eV loss (our Figure 5b(iii)), whereas all RIXS spectra of charged LNMO by Massel *et al.* are quite broad. Our Jahn-Teller distorted $S = \frac{1}{2}$ species is calculated to have a broad loss spectrum, and the neighboring Mn octahedra in LNMO are likely to change the local geometry around the Ni towards longer bond lengths versus in the layered LiNiO_2 .

Overall, it is interesting but quite challenging to interpret the spectra from Massel *et al.* due to the off-stoichiometry of their materials. It is further interesting that their reference spectrum for K_3NiF_6 is completely distinct from the half-charge spectra of LNMO and from our $S = \frac{1}{2}$ spectrum. Ni may very well be disproportionated in K_3NiF_6 as in rare-earth nickelate perovskites below their insulator-to-metal transitions.

R2: Summary: Poletayev et al. uses DFT and DFT-based molecular dynamic simulations to obtain an atomic and electronic structural model for $LiNiO_2$ at high and low temperatures. This model is not the thermodynamically most stable one (which is a few meV/atoms lower) and is dependent on the calculation parameters (in particular the functional). The obtained model is composed of bond and charge (spin) disproportionated Ni sites with dynamic exchange between Ni sites. The theoretical thermodynamic phase obtained from relaxing the high symmetry rhombohedral phase contains Jahn-Teller distorted Ni sites. Poletayev et al. also report experimental Ni L-edge spectra, namely variable temperature XAS, XMCD and RIXS with variable excitation energy. Single cluster ligand field multiplet calculations based on bond/charge disproportionated model reported by the authors allows a decent reproduction of these Ni L-edge spectroscopies. Finally, Poletayev et al. discuss the reported model with regard to previously reported electronic conductivity and Li/Ni antisite defects in $LiNiO_2$. $LiNiO_2$ conductivity is thermally activated and higher than JT distorted $NaNiO_2$, which the authors attribute to polarons hopping in the bond/charge disproportionated $LiNiO_2$ system. Ni^{2+} antisite mixing is proposed to be facilitated by the presence of Ni^{2+} in the charge disproportionated $LiNiO_2$.

General comment: Considering the amount of recent work dedicated to understand $LiNiO_2$ local and electronic structure, I believe this topic is of interest to the battery/physics community. The novelty of Poletayev et al. work is nested into (1) the dynamic nature of the charge/bond disproportionation, (2) new interesting spectroscopic datasets (variable temperature Ni L-edge XAS, XMCD and variable excitation energy Ni L-edge RIXS, (3) multiplet calculations of XMCD and RIXS and comparison with the experimental data. Note that the static bond/charge disproportionation model have been reported before [10.1103/PhysRevB.100.165104], while the dynamic nature of $LiNiO_2$ postulated by Sicolo et al. [10.1021/acs.chemmater.0c03442]. In terms of datasets, some XAS and RIXS were reported by Jacquet et al. [10.1002/aenm.202401413]. I believe that the modelling, spectroscopic simulations, data acquisition and analysis is well done and generally close to state-of-the-art.

We thank the reviewer for their reading of our manuscript. We note that neither the computational ground state ($P2_1/c$) nor the $P2/c$ structure have ever been experimentally observed at any temperature, and partial disproportionation contributes additional stabilization from configurational entropy relative to $P2_1/c$ (all Ni^{3+}) or $P2/c$ (all ordered $Ni^{2+/4+}$) computed phases. We have cited Foyevtsova *et al.* and Sicolo *et al.* (refs. 21 and 19,

respectively) and credited both in our introduction. To our knowledge, neither the dynamic interconversion between Ni^{3+} and $\text{Ni}^{2+/4+}$ nor its temperature evolution have been previously proposed or reported. Further, Jacquet *et al.* [6] cite a pre-print of the present work, and their manuscript appeared (chemRxiv Mar 27, 2024) after our model was already applied to interpret x-ray Raman spectra in ref. 36 (chemRxiv Mar 20, 2024).

R2: Questions: (1) Do I understand correctly that the disproportionated model is obtained by relaxing a starting structural model which is already disproportionated? As suggested by “When this asymptotic threefold disproportionated structure is relaxed ...” in the supplementary information? Along the same line, the paragraph starting by “The limiting case of the three-state system” is unclear because it mixes theoretical results and literature. Clarifying how the model is obtained in the main text would be welcome (even if it is not the most stable system, as the authors mention theoretical calculations on LiNiO_2 a correlated, locally distorted system potentially dynamic might be the limit of current theories).

We are happy to clarify. The discussion in the supplementary information referenced by the reviewer indeed describes how we arrived at the idea of considering a three-fold disproportionated structure like that of Foyevtsova *et al.*: it is similar to other three-fold motifs in hexagonal planar spin lattices and to mixed (Ni,Mn,Co) oxide cathodes. We have already included these analogies in the main text: “We note the similarity between this limiting structure and the three transition-metal sublattices in $\text{Li}(\text{NiMnCo})\text{O}_2$ [ref], noncollinear spin models for hexagonal lattices [ref], and the disproportionated structure of AgNiO_2 [refs].”

Since there is at least a kinetic barrier to the interconversion between three Ni states (Figure 2ab), any DFT relaxation that needs to change spin state encounters saddle points along its path, which drastically slows down the calculations. To simplify the relaxation of the asymptotic threefold ordered structure, and the relaxation of many training structures for the cluster expansion, we pre-distorted local Ni environments by increasing the volume for $S = 1$, decreasing the volume for $S = 0$, and adding a Jahn-Teller distortion for $S = \frac{1}{2}$ while preserving volume. We have already noted this in the methods.

The possible confusion in the main text has likely arisen because we take care to not claim that this single ordered threefold structure, or indeed any single structure, represents LiNiO_2 at practical temperatures (300 K). This is due to the continuous interconversion of Ni species and to the incremental changes in the fractions of the three Ni species with temperature. We use the “limiting” and “asymptotic” qualifiers to alert readers to the fact that no single ordered structure accurately represents the dynamic nature of LiNiO_2 at practical temperatures. More rigorously, we opt for a statistical description based on grand-canonical Monte-Carlo simulations with a fitted cluster expansion lattice Hamiltonian (Figure 2d and Methods). In this picture, the material is characterised by the temperature-dependent fractions of the three Ni species. We use the ab initio molecular dynamics (Figure 2ab) to inform experiments verifying the temperature dependence of Ni specie populations. We

understand the possible confusion, since most computational studies focus on singular structures. We take this approach to avoid over-interpreting computational predictions given their sensitivity to the level of theory, which is noted by the reviewer. Indeed, the first pre-print of this study posted to arXiv in November 2022 prior to experimental verification and subsequently cited by Jacquet *et al.* reflects our own surprise at the disagreement between simulations that gave rise to our experimental design.

As an aside, a DFT relaxation from the average R-3m structure with equal Ni–O bond lengths is very cumbersome due to the metallic nature of the starting structure and the appearance of a band gap due to the distortions of the bond lengths. Since the treatment of the metal requires very small electronic convergence steps, whereas once the band gap is established such steps lead to prohibitively long calculation timelines. As such, a change of convergence algorithm is necessary once the band gap is reached. In such relaxations, we have observed partial disproportionation, e.g., two to four Ni ions could disproportionate within a layer of nine, with the rest remaining as $S = \frac{1}{2}$. We could not identify any pattern to such structures and saw no reason to treat them as unique or final for the purpose of comparisons to experimental data (diffraction, EXAFS, and so on). As noted above, a simple relaxation may not have overcome all barriers identified from molecular dynamics. It was therefore logical to examine a limiting or asymptotic structure for an initial comparison to experiments.

R2: (2) The variable temperature Ni L-edge IPFY is interesting (especially the use of IPFY which is currently the best way to measure intensities in fluorescence mode, to the best of my understanding). However, the changes with temperature are really small. $S = \frac{1}{2}$ specie phase fraction only changes from 35% to 37% heating from 25 K to 300 K. That's in the experimental error if we consider that the reproducibility of the phase fraction measurement is 2% (the reproducibility between two measurements as mentioned by the authors in the figure caption of extended figure 1). Can the author provide a more accurate reproducibility? Otherwise, I'm afraid to say that the temperature change in $S = 0$, $S = \frac{1}{2}$ and $S = 1$ population is not experimentally supported.

The temperature dependence of spectra in Figure 2c is visible by the naked eye even before any quantitative fitting (which is shown in Extended Data Figure 1) and is first discussed qualitatively in the main text. We further note that the increase in $S = \frac{1}{2}$ population between 20 K and 375 K was reported as 6%, significantly above the error thresholds of our estimates.

In principle, it is possible to include an arbitrarily large number of parameters in fitting spectra, such as component-specific and temperature-specific spectral broadening, background terms, and so on. Since interconversion between three Ni species at elevated temperature is more rapid, the central region of the L_3 edge becomes flatter than otherwise, which could be separately accounted for with more complex spectral shapes. To reduce the number of fitting parameters, we use a double arctan background over the $L_{3,2}$ edges. The background varies between 0.7-1.1% of the edge height at the L_3 edge, i.e., it remains essentially flat across the fitting area, and the sensitivity of fitting to it is low, so we fix it here. Below (Figure R1) we show residuals to least-squares fitting over the L_3 edge (two free

parameters since three concentrations add up to 100%), having fixed the background and broadening.

Figure R1: Least-squares fitting to Ni species populations over the L_3 edge of temperature-dependent IPFY spectra. From top to bottom: 375 K, 300 K, and 20 K.

The resulting percentages of $S = \frac{1}{2}$ species are $34.0 \pm 0.3\%$ at 20 K, $37.7 \pm 0.3\%$ at 300 K, and $41.0 \pm 0.5\%$ at 375 K. Indeed, our initial fitting was less accurate than this, but this result falls well within our intentionally conservative 2% error estimate. The least-squares fitting only strengthens our initial conclusions. The slight evolution of spectral shapes with heating, for which our model does not account, manifests in an apparent decrease in Ni_{Li} concentration with heating, which is the difference between $S = 1$ and $S = 0$ fractions, from $3.6 \pm 0.3\%$ at 20 K to $2.3 \pm 0.3\%$ at 300 K and $2.0 \pm 0.4\%$ at 375 K. This decrease is fictitious: the low-temperature estimate is the more accurate one. That the $S = 1$ feature, but not the $S = 0$ feature, appears sharper in the model versus experiment at 20 K and 300 K is consistent with our proposed mechanism of charge transport as exchange predominantly between the $S = 1$ and $S = \frac{1}{2}$ species, which should selectively broaden their spectra but leave the $S = 0$ feature intact until elevated temperatures. We amend our estimated fractions of the $S = \frac{1}{2}$ Ni species as 34%, 38%, and 41%, update Extended Data Figure 1, and retain our conservative error estimate as 2%.

Another measure of reproducibility is repeat measurements. In the main text, we have averaged two sets of scans at 300 K: taken before and after cooling to 20 K. Below (Figure R2) we show dis-aggregated fitting of the two sets of scans at 300 K. The percentages of $S = \frac{1}{2}$ species from least-squares fitting are $37.3 \pm 0.4\%$ for the first set of scans (Figure R2, top) and

37.6±0.3% for the second (Figure R2, bottom). These too are well within the error bounds we have reported. The Ni_{Li} concentrations are 3.5±0.4% and 1.8±0.3%, respectively.

Figure R2: Least-squares fitting Ni species populations over the L₃ edge of temperature-dependent IPFY spectra. From top to bottom: initial scans at 300 K and scans at 300 K following cooling to 20 K.

R2: (3) The statement “a key novelty of our work is the confirmation that these formally high valence species are present in the pristine, fully lithiated material” is debatable. Indeed, other works have made similar observations recently [10.1021/acsnenergylett.4c00360, 10.1002/aenm.202401413]

We note that Jacquet *et al.* [6] cite a pre-print of the present work as their ref. 59, and that the study by Jacquet *et al.* was first posted on Mar 27, 2024, whereas our study in (ref. 36, now peer-reviewed [1]) that applies the model that is detailed here was posted on Mar 20, 2024. We reproduce below (Figure R3) the percentages of Ni species from XRS in ref. 36, fitted using the model in the present work, and the evolution of Ni species populations modelled by the cluster expansion here. These show excellent agreement up to the fraction of Ni_{Li} defects that remain reduced in the delithiated material (the cluster expansion does not yet account for these defects).

We note that the reduced surface layer in charged LiNiO₂ (see our ref. 36 or [1]) extends beyond the probing depth of even the highest energy of HAXPES. This means that HAXPES spectra of even the fully delithiated samples possess a substantial fraction of S = 1 (formally Ni²⁺) species at all probing depths. Therefore, no XPS reference spectra can correspond to pure Ni species, and additional care should be taken to verify the correspondence of HAXPES results to bulk Li_xNiO₂.

Figure R3: Ni specie populations fitted to x-ray Raman spectra (left, ref. 36) using the ligand-field multiplet model in the present work and from grand canonical Monte-Carlo simulations (this work, right).

R2: (4) In the RIXS interpretation, I find difficult to claim “the fluorescence feature [...] extends to < 1 eV” without performing a decomposition of the RIXS signal into “Raman-like” and ‘fluorescence-like’ signal before as Bisogni et al. performed for example [10.1038/ncomms13017]. Currently, this statement is merely based on visual observation.

We note that the decomposition of loss spectra by Bisogni *et al.* is somewhat empirical: they fit Gaussians to each loss spectrum. This is possible because they only use one set of Gaussians for d-d transitions at all energies. Here, because there are many more d-d transitions peaking at distinct excitation energies and overlapping with multiple sets of broad charge-transfer states, this is not feasible. The fluorescence line is evident at 1-3 eV at 852.5-854.5 eV excitation energies, respectively (Figure 5b), and Figure S5 shows predicted RIXS spectra for three Ni species. Below (Figure R4) we show the spectra at 852.0 eV and 851.5 eV; the expected positions of the fluorescence line are 0.5 eV loss, and 0 eV loss, respectively, but it is clear within the noise of the measurement that fluorescence does not reach the elastic line, as it would for a metal (see for example Figure 5 of Bisogni *et al.*). At all energies, the measured intensity (blue) between 0 eV loss and 1 eV loss reaches zero, in line with the text of our manuscript. We interpret the low-loss shoulder of the main 1 eV peak at 852.0 eV, which is absent at 852.5 eV, as the fluorescence, since no d-d transitions are predicted at lower loss than the main $S = 1$ peak (1.2-1.3 eV model, an over-estimate of 1 eV experiment).

Figure R4: Loss spectra at excitation energies 851.5 eV (top), 852.0 eV (middle), and 852.5 eV (bottom, reproducing Figure 5b(iii)).

(5) Why would the “gradual decrease in this unit cell distortion with heating” be consistent with “gradually increasing proportion of $S = \frac{1}{2}$ species” ? NaNiO_2 has plenty of $S = \frac{1}{2}$ species but is distorted.

The simple difference is that the distortion in NaNiO_2 is collective, i.e., all long Jahn-Teller axes point in the same direction and give rise to the overall unit cell distortion. The long axes of $S = \frac{1}{2}$ species in LiNiO_2 point in random directions and cannot give rise to an appreciable overall distortion of the unit cell. This is because they are constantly destroyed and re-created via dis- and comproportionation, respectively.

R2: (6) How did the authors obtain the fraction of d^7 , d^8L , d^9L^2 ... configuration shown in extended Figure 2. Also, the bond lengths are mentioned in the figure caption but not shown in the figure.

The configurations are obtained from ligand-field calculations parametrized from the bond lengths obtained from DFT as detailed in the methods. The wavefunctions are projected on the corresponding d^xL^y basis states and the norm of the projection is computed in Quancy. For example, if the projection operator for the d^xL^y basis state is O_{xy} and the ground state wavefunction is $|\psi_0\rangle$, the calculation is $\langle \psi_0 | O_{xy}^\dagger O_{xy} | \psi_0 \rangle$. We have included a clarification in the methods: “To obtain the d^xL^y terms for the ground-state configurations, the wavefunctions are projected onto the corresponding basis set in Quancy.”

We have now added the bond lengths to Extended Data Figure 2 and are thankful to the reviewer for pointing out this omission.

R2: (7) typo in the energy resolution of the XAS (70 eV...)

The resolution of the fluorescence detector is 70 eV, sufficient to distinguish elemental fluorescence lines.

R2: (8) Electrodes has been measured for temperature dependent XMCD but there formulation/preparation is not described.

The electrode preparation is as follows: mixtures of 80 wt% LiNiO₂, 10% wt% acetylene black, and 10 wt% polytetrafluoroethylene (PTFE) binder were calendared for free-standing electrodes. We have amended the methods to include this: “*to prepare free-standing electrodes for spectroscopic measurements, LiNiO₂ powder was mixed with acetylene black and polytetrafluoroethylene (PTFE) as binder in weight ratios 80:10:10, and calendared*”.

R2: (9) Please include the energy resolution for the RIXS measurement

The energy resolution of the RIXS measurement at beamline I21 of Diamond Light Source was ≤60 meV at 850 eV. This is now included in the methods section.

**R2: (10) Regarding the multiplet calculations, I wonder why the authors didn't use double site multiplet calculations has performed for the nickelates?
[10.1103/PhysRevX.8.031014]**

Double-cluster multiplet calculations are helpful to accurately represent the ground state of perovskite nickelates at temperatures below the insulator-to-metal transition; see, for example, refs. 22 and 37 in the main text. Two considerations informed the use of single clusters here. First, to represent the three Ni species in LiNiO₂, a triple cluster would be required, which is computationally prohibitive. Second, one of us (R.J.G.) has shown that if the ground-state electron numbers can be approximated, as was done here from density-functional theory geometries, then single-cluster calculations can be sufficient [7]. This is due to the weaker intercluster interactions by the core hole in the final state [7].

R3: This is a fascinating and impressive piece of work which brings together a range of state-of-the-art computational and experimental techniques to probe the electronic structure of LiNiO₂ - a material of intrinsic interest and importance owing to its applications in battery technology. The present referee has most expertise in the computational techniques employed and can confirm that these are appropriate and have been carefully employed. The results demonstrating dynamic disproportionation are particularly interesting and novel and have wider implications for our understanding of the electronic structure of transition metal oxides. The work is of the quality and

originality to justify publication in Nature Comms. I have two recommendations for minor changes:

We are very grateful to the reviewer for their time and encouraging comments!

1. The authors comments that the system has a high degree of covalence. Could they for at least some configurations provide a Bader charge analysis, which would be helpful additional information.

Partial charges could arise not only from covalence, but also from electron or hole delocalization, and we have avoided such terms so far. We have avoided these terms to focus on explaining experimental observables. The Bader charges for the Ni species in the asymptotic threefold disproportionated structure of LiNiO_2 are 1.40, 1.53, and 1.66, and Bader charges for the oxygen ions vary between 1.24-1.29. In delithiated O3-stacked NiO_2 , the Bader charges are 1.68 and 0.84 for Ni and O, respectively. There has been considerable debate over the many methods for determining partial charges, for example [8,9]. Another measure of ionicity is the percentage of purely ionic d^{6-8} terms in the results of ligand-field multiplet calculations. For $S = 1$ (formally Ni^{2+}) we cite 80% d^8 , for $S = \frac{1}{2}$ (formally Ni^{3+}) we cite 25% d^7 , and for $S = 0$ (formally Ni^{4+}) we cite 6% d^6 . We use spins and formal oxidation states as a basis set to distinguish local Ni environments. Spin has been successfully used to distinguish transition-metal species, especially for constructing cluster expansion Hamiltonians as done here [4,5]. We include the Bader charges in the supplementary information next to the discussion of geometry relaxations: “*Bader charges in threefold disproportionated LiNiO_2 are 1.40, 1.53, and 1.66 for Ni, and vary between 1.24-1.29 for O. In O3-stacked NiO_2 , the Bader charges are 1.68 and 0.84 for Ni and O, respectively.*”

2. Can they clarify whether the DFT calculations were performed on the unit cell or on a supercell.

Ab initio molecular dynamics have used 108-atom cells (3 x 3 supercells of the conventional three-layer rhombohedral cell), and the cluster expansion was constructed from supercells between 48-144 atoms at full lithiation. Our use of these large supercells allows the system to disproportionate if that is preferred, and to avoid over-constraining the relaxation with artificially small cells. The simplest unit cell of the asymptotic threefold disproportionated structure is a $\sqrt{3} \times \sqrt{3}$ supercell of the conventional unit cell, with a 30-degree rotation. We plan to include reference disproportionated structures with the manuscript. We have amended the methods to clarify this point (italics highlight the addition): “Reference structures for training were chosen to be large enough to allow for disproportionation should that be favorable (*4-12 Ni ions per layer, 48-144 atoms*)”

References:

- [1] L. An, J. E. N. Swallow, P. Cong, R. Zhang, A. D. Poletayev, E. Björklund, P. N. Didwal, M. W. Fraser, L. A. H. Jones, C. M. E. Phelan, N. Ramesh, G. Harris, C. J. Sahle, P.

- Ferrer, D. C. Grinter, P. Bencok, S. Hayama, M. S. Islam, R. House, P. D. Nellist, R. J. Green, R. J. Nicholls, and R. S. Weatherup, *Energy Environ Sci* **17**, 8379 (2024).
- [2] M. Juelsholt, J. Chen, M. A. Pérez-Osorio, G. J. Rees, S. De Sousa Coutinho, H. E. Maynard-Casely, J. Liu, M. Everett, S. Agrestini, M. Garcia-Fernandez, K. J. Zhou, R. A. House, and P. G. Bruce, *Energy Environ Sci* **17**, 2530 (2024).
- [3] K. McColl, A. D. Poletayev, M. S. Islam, and B. J. Morgan, *ChemRxiv:10.26434/Chemrxiv-2024-Qs91t* (2024).
- [4] J. H. Yang, T. Chen, L. Barroso-Luque, Z. Jadidi, and G. Ceder, *NPJ Comput Mater* **8**, 133 (2022).
- [5] L. Barroso-Luque, J. H. Yang, F. Xie, T. Chen, R. L. Kam, Z. Jadidi, P. Zhong, and G. Ceder, *J Open Source Softw* **7**, 4504 (2022).
- [6] Q. Jacquet, N. Mozhzhukhina, P. N. O. Gillespie, G. Wittmann, L. P. Ramirez, F. G. Capone, J.-P. Rueff, S. Belin, R. Dedryvère, L. Stievano, A. Matic, E. Suard, N. B. Brookes, A. Longo, D. Prezzi, S. Lyonnard, and A. Iadecola, **2401413**, 1 (2024).
- [7] J. Li, R. J. Green, C. Domínguez, A. Levitan, Y. Tseng, S. Catalano, J. Fowlie, R. Sutarto, F. Rodolakis, L. Korol, J. L. McChesney, J. W. Freeland, D. Van der Marel, M. Gibert, and R. Comin, *Nat Commun* **15**, 7427 (2024).
- [8] M. Jansen and U. Wedig, *Angewandte Chemie - International Edition* **47**, 10026 (2008).
- [9] A. Walsh, A. A. Sokol, J. Buckeridge, D. O. Scanlon, and C. R. A. Catlow, *Nat Mater* **17**, 958 (2018).